# R2Q: RESIDUAL REFINEMENT QUANTIZATION FOR RObust 2-Bit Large Language Models

## ABSTRACT

The dramatic growth of Large Language Models (LLMs) has been accompanied by significant computational and memory demands, driving the adoption of low-bit quantization. While 8-bit and 4-bit formats have become standard, ultra-low-bit quantization, particularly 2-bit, presents a substantial challenge due to severe accuracy degradation. To address this, we propose Residual Refinement Quantization (R2Q)—a novel 2-bit quantization strategy that decomposes the quantization process into two sequential 1-bit subproblems, enabling adaptive quantization lattice. Extensive experiments on Llama, OPT and Qwen were conducted across diverse benchmarks, including question answering, commonsense reasoning, and language modeling. The results demonstrate that R2Q consistently outperforms state-of-the-art 2-bit quantization baselines in both coarse-grained and fine-grained settings. The refinement-based design of R2Q not only enhances quantization performance but also improves training stability and convergence under aggressive compression. Furthermore, R2Q is modular by design and can be seamlessly integrated into existing quantization-aware training (QAT) pipelines.

## 1 INTRODUCTION

Large Language Models (LLMs) have brought transformative changes to natural language processing, demonstrating exceptional performance across tasks ranging from text generation to complex reasoning. Yet, their massive scale and computational demands remain a major obstacle. To mitigate this, the community has progressively adopted more efficient data formats—beginning with FP16 (Micikevicius et al., 2017) and BFloat16 (BF16) (Kalamkar et al., 2019)—and now increasingly relies on quantization, particularly 8-bit and 4-bit representations. Quantization methods, including integer (Dettmers et al., 2022; Yao et al., 2022; Xi et al., 2023), floating-point (Micikevicius et al., 2022; Liu et al., 2023a), and NormalFloat (Dettmers et al., 2023), have proven effective in reducing memory and compute costs while largely preserving model accuracy.

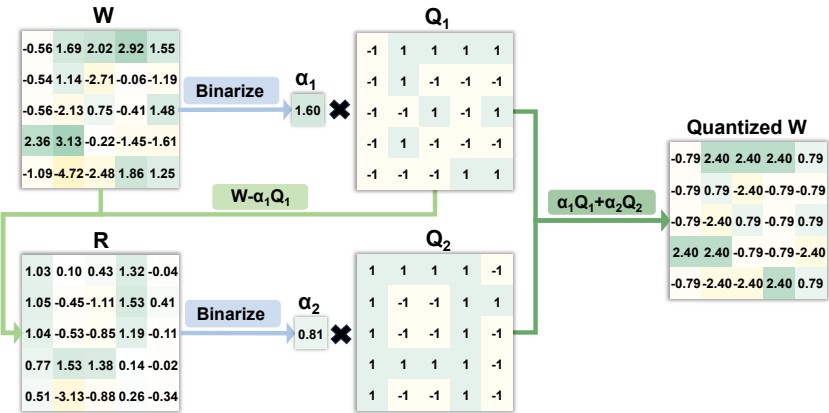

Figure 1: Residual Refinement Quantization (R2Q). Full-precision weight $\mathbf{W}$ and its first-step residual $\mathbf{R}$ are binarized into two 1-bit kernels, $\mathbf{Q}_1$ and $\mathbf{Q}_2$, which are combined to reconstruct $\mathbf{W}$.

However, resource-limited devices such as smartphones, UAVs (Mohsan et al., 2023), and AR/VR headsets (Arena et al., 2022; Meta Platforms, Inc., 2025) impose stricter constraints. For instance, a 7B-parameter LLM quantized to 4-bit still requires 3.5 GB for weights alone. On a smartphone with 8 GB RAM, this leaves little headroom after accounting for the KV cache ($\approx 2$ GB), the operating system ($\approx 2$ GB), and other applications. Such pressure underscores the need for more aggressive compression, motivating research into 2-bit, ternary, and even binary quantization to enable on-device applications like offline translation and voice assistants.

Binary quantization represents the extreme case, eliminating multiplications altogether (Rastegari et al., 2016), but typically requires retraining from scratch (Wang et al., 2023) or extensive fine-tuning (Xu et al., 2024) to remain competitive. Ternary quantization (Chen et al., 2024; Ma et al., 2024; Yuan et al., 2024) improves efficiency by introducing zero, yet remains hindered by binary-oriented hardware. In contrast, **2-bit quantization** strikes a more practical balance, offering strong compression without the drawbacks of ternary or binary schemes. A key challenge, however, lies in the absence of effective mapping strategies. Existing methods (Chee et al., 2023; Du et al., 2024; Shao et al., 2023) typically adopt integer quantization, where high-precision values are scaled, shifted, and rounded (RTN). Given the limited expressiveness of 2-bit formats, performance preservation often requires fine-grained grouping (Shao et al., 2023; Du et al., 2024).

To overcome the limitations of coarse-grained 2-bit quantization, we introduce **Residual Refinement Quantization (R2Q)**. As shown in Figure 1, R2Q decomposes 2-bit quantization into two 1-bit subproblems: the first provides a coarse approximation, while the second refines the residual. This yields an **adaptive** quantization lattice, allowing R2Q to better capture weight distributions. Unlike RTN's static lattice, R2Q achieves more accurate parameter mapping (Figure 2), resulting in a **stable and convergent** Quantization-Aware Training (QAT). Furthermore, representing weights with two 1-bit kernels preserves the **matmul-free** efficiency of binary and ternary quantization—an advantage absent in other 2-bit schemes. Finally, R2Q's **modular design** enables seamless integration into existing QAT pipelines. Our contributions are summarized as follows:

- We propose R2Q, a 2-bit quantization strategy that decomposes the problem for LLMs into two simpler 1-bit sub-problems, enabling a more adaptive quantization lattice.

- We designed R2Q as a modular solution readily integrated into existing QAT frameworks, offering substantial improvements in stability and convergence efficiency.

- We validate R2Q through extensive benchmark experiments, showing it outperforms existing methods under 2-bit quantization, especially with coarse-grained group quantization.

The rest of the paper is organized as follows: Section 2 reviews related work. Section 3 introduces preliminaries. Section 4 presents the R2Q framework. Section 5 conducts experiments. Section 6 discusses results and future directions.

## 2 RELATED WORK

This section reviews related work on quantization techniques for LLMs, which can be broadly categorized along two dimensions: *quantization format* and *quantization strategy*.

**Quantization Format.** A straightforward approach to model compression is reducing fractional and exponent precision, as in FP16 and BF16. BF16, with improved numerical stability, is now standard in state-of-the-art models such as OpenPangu, Llama (Dubey et al., 2024), and Qwen (Bai et al., 2023). However, growing model scales demand more aggressive quantization. At its core, LLM quantization maps continuous values to a finite discrete set for efficient compression.

For 8-bit formats, 8-bit integer (INT8) and 8-bit floating-point (FP8) are prominent. LLM.int8() (Dettmers et al., 2022) successfully implemented INT8 quantization for LLMs by using vector-wise quantization and a special mixed-precision decomposition. Concurrently, FP8 (Micikevicius et al., 2022), such as E4M3 and E5M2, were proven effective for both training and inference by matching the performance of 16-bit formats without requiring hyperparameter changes.

In the 4-bit regime, INT4 methods such as Xi et al. (2023) mitigate activation outliers via Hadamard transforms and importance sampling. FP4 (Liu et al., 2023a) improves robustness with optimized

quantization parameters and exponent shifting. QLoRA (Dettmers et al., 2023) further enables efficient fine-tuning by introducing the NormalFloat4 (NF4) format with LoRA and double quantization.

Binary and ternary quantization commonly rely on the indicator (Wang et al., 2023; Xu et al., 2024) and sign (Chen et al., 2024; Ma et al., 2024; Yuan et al., 2024) functions, respectively. The optimality of these mappings has been established for certain distributions (Rastegari et al., 2016; Li et al., 2016). Current 2-bit methods typically use integer-based formats with RTN (Chee et al., 2023; Du et al., 2024; Shao et al., 2023), inherited from higher-bit quantization, but these are ill-suited to the limited representational capacity of four discrete values.

**Quantization Strategies.** Quantization strategies can be grouped into two primary categories: Post-Training Quantization (PTQ) and Quantization-Aware Training (QAT).

PTQ applies quantization without retraining, often using calibration data. GPTQ (Frantar et al., 2022) leverages second-order approximations of weight sensitivity via the Hessian matrix to achieve accurate 3/4-bit quantization. AWQ (Lin et al., 2024) and SmoothQuant (Xiao et al., 2023) focus on activation-aware weight scaling to preserve salient channel representations. QuaRot (Ashkboos et al., 2024) and FlatQuant (Sun et al., 2024) simplify quantization by flattening weight or activation distributions. While efficient, PTQ methods degrade sharply under extreme bit-widths (e.g., 2-bit).

QAT integrates quantization into training or fine-tuning. Although more resource-intensive, QAT delivers superior results in ultra-low-bit settings. QLoRA (Dettmers et al., 2023) combines 4-bit quantization with LoRA, leveraging NF4 and double quantization to minimize memory usage. LLM-QAT (Liu et al., 2023b) employs data-free distillation from a full-precision teacher, while BitDistiller (Du et al., 2024) introduces a confidence-aware KL divergence objective tailored for 2-bit quantization. Lee et al. (2025) propose a progressive approach that first applies PTQ to obtain a 4-bit model, followed by QAT with knowledge distillation to further reduce precision to 2 bits. Following this line of work, we adopt a distillation-based QAT framework.

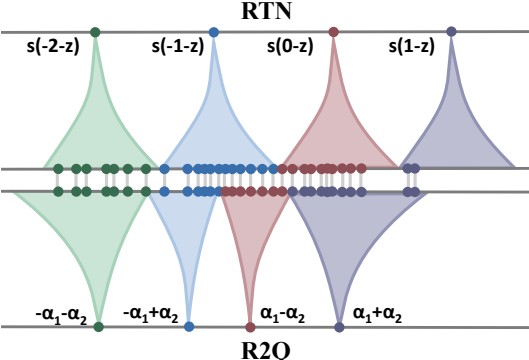

Figure 2: Comparison of RTN (top) and R2Q (bottom) in 2-bit quantization. Gray lines connect points with the same real value. Points within the cone base are mapped to the apex, forming a **quantization lattice**. RTN relies on scaling factor $s$ and zero point $z$, enforcing uniform allocation. In contrast, R2Q employs two kernel scales $\alpha_1$ and $\alpha_2$, enabling **adaptive** mapping for imbalanced distributions. As shown in the top figure, RTN wastes levels—only two values occupy the purple lattice, leaving four quantization levels underutilized.

## 3 PRELIMINARIES

This section introduces the preliminaries for LLM quantization, which covers quantization and dequantization, group-wise quantization and static lattice mapping.

**Problem Formulation.** LLMs are predominantly decoder-based Transformers, which have substantial memory and computational costs largely due to their linear layers. This makes quantization a crucial technique for efficient deployment. We focus our analysis on a single linear layer, as the process applies to all others. The operation is defined as

$$f(\mathbf{X}) = \mathbf{W}x, \tag{1}$$

where $\boldsymbol{x} \in \mathbb{R}^{D_{in}}$ is the input vector and $\mathbf{W} \in \mathbb{R}^{D_{out} \times D_{in}}$ is the full-precision weight matrix.

Quantization maps the continuous values in $\mathbf{W}$ to a discrete set of values from a predefined codebook $\mathcal{C} = \{c_1, c_2, \ldots, c_{2^k}\}$, where $k$ is the bit-width. This mapping can be performed by function

$$\mathbf{Q} = \mathcal{Q}_\theta(\mathbf{W}). \tag{2}$$

Here, $\mathbf{Q}$ is the quantized weight with each entry $Q_{ij} \in \mathcal{C}$. $\mathcal{Q}_\theta$ denotes the quantization fuction. The parameters $\theta$ (e.g., scaling factors, zero-points) are used for both quantization and dequantization. An approximation of the original weights $\hat{\mathbf{W}}$ is recovered via a dequantization function $\mathcal{D}_\theta$, which is

$$\hat{\mathbf{W}} = \mathcal{D}_\theta(\mathbf{Q}) \approx \mathbf{W}. \tag{3}$$

**Group-Wise Quantization.** We use group-wise quantization, a technique common in most quantization methods (Frantar et al., 2022; Shao et al., 2023; Liu et al., 2023b; Du et al., 2024) to achieve improved accuracy. This approach partitions the matrix $\mathbf{W}$ into smaller and disjoint groups, making quantziation operate at a finer granularity. Specifically, $\mathbf{W}$ is divided into $N$ groups,

$$\mathbf{W} = [\boldsymbol{w}^{(1)}, \boldsymbol{w}^{(2)}, \ldots, \boldsymbol{w}^{(N)}], \quad \boldsymbol{w}^{(i)} \in \mathbb{R}^G, \tag{4}$$

where $G$ is the group size, and each vector $\boldsymbol{w}^{(i)}$ is a contiguous subset of weights. For each group, we quantize and dequantize it independently using a group-specific function

$$\hat{\boldsymbol{w}}^{(i)} = \mathcal{D}_{\theta^{(i)}} \left( \mathcal{Q}_{\theta^{(i)}} \left( \boldsymbol{w}^{(i)} \right) \right), \quad \hat{\boldsymbol{w}}^{(i)} \in \mathcal{C}^G, \tag{5}$$

The quantized matrix $\hat{\mathbf{W}}$ is then reconstructed by concatenating the quantized groups

$$\hat{\mathbf{W}} = [\hat{\boldsymbol{w}}^{(1)}, \hat{\boldsymbol{w}}^{(2)}, \ldots, \hat{\boldsymbol{w}}^{(N)}]. \tag{6}$$

This group-wise strategy enables finer adaptation to local weight distributions and mitigates quantization errors, particularly in low-bit quantization scenarios.

**Static Lattice Mapping.** Round-to-Nearest (RTN), which employs a static lattice, is a common strategy used in 2-bit quantization (Du et al., 2024; Chee et al., 2023). This method simply scales and shifts a fixed set of integer base points, preserving the uniform spacing between quantization levels.

In RTN, an affine transformation maps real-valued weights to a signed or unsigned integer range. Taking the signed case as an example, the target range is $[q_{\min}, q_{\max}] = [-(2^{k-1}), 2^{k-1} - 1]$. For a given group of weights with real-valued range $r_{\min}^{(i)} = \min(\boldsymbol{w}^{(i)})$ and $r_{\max}^{(i)} = \max(\boldsymbol{w}^{(i)})$, the quantization process is defined by a scaling factor $s$ and a zero-point $z$, which can be fomulated as

$$s^{(i)} = \frac{r_{\max}^{(i)} - r_{\min}^{(i)}}{q_{\max} - q_{\min}},$$

$$z^{(i)} = \left\lfloor \frac{q_{\min} - r_{\min}^{(i)}}{s^{(i)}} \right\rceil, \tag{7}$$

$$\boldsymbol{q}^{(i)} = \mathcal{Q}_{s^{(i)}, z^{(i)}} \left( \boldsymbol{w}^{(i)} \right) = \text{clip} \left( \left\lfloor \frac{\boldsymbol{w}^{(i)}}{s^{(i)}} \right\rceil + z^{(i)}, q_{\min}, q_{\max} \right),$$

where $\lfloor \cdot \rceil$ means rounding to the nearest integer and $\text{clip}(\cdot)$ ensures that the quantized values remain in the target range. The dequantized weight $\hat{\boldsymbol{w}}^{(i)}$ is then recovered by,

$$\hat{\boldsymbol{w}}^{(i)} = \mathcal{D}_{s^{(i)}, z^{(i)}} \left( \boldsymbol{q}^{(i)} \right) = s^{(i)} \cdot (\boldsymbol{q}^{(i)} - z^{(i)}). \tag{8}$$

While effective for 4-bit or higher quantization with enough mapping points to handle non-uniform distributions Dettmers et al. (2022); Xi et al. (2023); Frantar et al. (2022), RTN's uniform lattice is ill-suited for 2-bit quantization. With only four levels, the non-uniform distribution of LLM weights clusters most values around central points, leaving a few outliers on others (Figure 2). This wastes limited quantization levels, reducing parameter distinctiveness. It therefore highlights the need for a more flexible lattice.

## 4   METHODOLOGY

Standard static lattice quantization, as discussed in Section 3, struggles under extremely low bit-width constraints (e.g., 2-bit). To mitigate this, we introduce R2Q, a novel 2-bit quantization scheme that decomposes the problem into two 1-bit subproblems. This section decomposes the 2-bit quantization problem, derives the optimal solution for 1-bit subproblems, and details the R2Q algorithm.

**2-bit Quantization via 1-bit Decomposition.** We begin by disaggregating the 2-bit quantiztaion problem. A 2-bit quantization codebook is limited to four values, i.e. $\mathcal{C} = \{c_1, c_2, c_3, c_4\}$. Directly optimizing a static lattice by uniformly shifting and scaling these values (as described in section 3) is non-trivial and often yields high approximation errors. We reformulate this problem by decomposing it into two 1-bit subproblems. Specifically, We decouple the 2-bit approximated vector $\boldsymbol{w}$ of $i$-th group by,

$$\boldsymbol{w}^{(i)} \approx \hat{\boldsymbol{w}}^{(i)} = \alpha_1^{(i)} \boldsymbol{q}_1^{(i)} + \alpha_2^{(i)} \boldsymbol{q}_2^{(i)}, \tag{9}$$

where $\boldsymbol{q}_1^{(i)}$ and $\boldsymbol{q}_2^{(i)}$ are vectors with elements in $\{+1, -1\}^G$. Here, the first kernel $\alpha_1^{(i)} \boldsymbol{q}_1^{(i)}$ serves as a coarse approximation of $\boldsymbol{w}^{(i)}$, while the second kernel $\alpha_2^{(i)} \boldsymbol{q}_2^{(i)}$ captures the residual for further refinement. The coefficients $\alpha_1^{(i)}$ and $\alpha_2^{(i)}$ are the corresponding scaling factors.

This decomposition creates a flexible codebook defined by $\{-\alpha_1^{(i)} - \alpha_2^{(i)}, \ -\alpha_1^{(i)} + \alpha_2^{(i)}, \ \alpha_1^{(i)} - \alpha_2^{(i)}, \ \alpha_1^{(i)} + \alpha_2^{(i)}\}$. The scaling factors $\alpha_1^{(i)}$ and $\alpha_2^{(i)}$ are adaptive and determine the importance of $\boldsymbol{q}_1^{(i)}$ and $\boldsymbol{q}_2^{(i)}$ thereby controlling the relative scale of the quantization lattice.

**Optimal Solution for the 1-bit Subproblem.** Consider an optimal binary approximation for a given vector $\boldsymbol{w}$, comprising a direction vector $\boldsymbol{q} \in -1, 1^G$ and a scaling parameter $\alpha \in \mathbb{R}^+$. This can be formulated as the constrained optimization problem, which seeks to minimize the Frobenius norm of the reconstruction error (Rastegari et al., 2016),

$$\alpha^*, \boldsymbol{q}^* = \underset{\alpha > 0, \mathbf{Q}}{\operatorname{argmin}} \|\boldsymbol{w} - \alpha \boldsymbol{q}\|_F^2, \quad \text{s.t.} \quad q_i \in \{-1, 1\}, \tag{10}$$

The objective function can be expanded by,

$$\|\boldsymbol{w} - \alpha \boldsymbol{q}\|_F^2 = \|\boldsymbol{w}\|_F^2 - 2\alpha \langle \boldsymbol{w}, \boldsymbol{q} \rangle_F + \alpha^2 \|\boldsymbol{q}\|_F^2, \tag{11}$$

where $\langle \cdot, \cdot \rangle_F$ is the Frobenius inner product. Since $q_i \in -1, 1$, it follows $q_j^2 = 1$, and thus $|\boldsymbol{q}|F^2 = \sum_i q_i^2 = G$. The objective becomes $|\boldsymbol{w}|F^2 - 2\alpha \sum_i w_i q_i + G\alpha^2$. We can solve for $\alpha$ and $\boldsymbol{q}$ iteratively. First, for fixed $\alpha > 0$, the objective is minimized by maximizing $\sum_i w_i q_i$. This maximum is achieved when $q_i$ shares the same sign as $w_i$. Therefore, the optimal solution for $\boldsymbol{q}$ is:

$$\boldsymbol{q}^* = \mathcal{H}(\boldsymbol{w}), \tag{12}$$

$$\mathcal{H}(w) = \begin{cases} +1 & w \geq 0, \\ -1 & w < 0, \end{cases} \tag{13}$$

where the $\mathcal{H}(\cdot)$ function is applied element-wise. Next, substituting $\boldsymbol{q}^*$ back into the objective and minimizing with respect to $\alpha$, we take the derivative of Eq. (11) and set it to zero

$$\frac{\partial}{\partial \alpha} \left( \|\boldsymbol{w}\|_F^2 - 2\alpha \langle \boldsymbol{w}, \boldsymbol{q}^* \rangle_F + G\alpha^2 \right) = -2\langle \boldsymbol{w}, \boldsymbol{q}^* \rangle_F + 2G\alpha = 0. \tag{14}$$

Solving for $\alpha$ yields the optimal scaling factor

$$\alpha^* = \frac{\langle \boldsymbol{w}, \boldsymbol{q}^* \rangle_F}{G} = \frac{\sum_i w_i \operatorname{sign}(w_i)}{G} = \frac{\sum_i |w_i|}{G} = \frac{\|\boldsymbol{w}\|_{\ell_1}}{G}, \tag{15}$$

where $\|\cdot\|_{\ell_1}$ denotes the $\ell_1$-norm. So far, we have derived the optimal solution to the 1-bit subproblem. Notably, it does not rely on specific weight distribution—a property not shared by ternary or higher-bit quantization (Li et al., 2016). This characteristic contributes to the improved robustness of R2Q.

**Residual Refinement Quantization.** Our quantization strategy employs a two-step strategy—the first to capture the coarse approximation, while the second to refine the residual—to decompose the 2-bit quantization problem. Formally, for each group of weights $\boldsymbol{w}^{(i)}$, we

1. Solve for $\boldsymbol{q}_1^{(i)}, \alpha_1^{(i)}$ by minimizing $\|\boldsymbol{w}^{(i)} - \alpha_1^{(i)}\boldsymbol{q}_1^{(i)}\|_F^2$,

2. Compute the approximation residual $\boldsymbol{r}^{(i)}$ and solve for $\boldsymbol{q}_2^{(i)}, \alpha_2^{(i)}$ by minimizing $\|\boldsymbol{r}_1^{(i)} - \alpha_2^{(i)}\boldsymbol{q}_2^{(i)}\|_F^2$.

First, a coarse 1-bit approximation is found by solving the optimization subproblem for the original weights $\boldsymbol{w}^{(i)}$, Using the optimal solutions derived in Eq. (12) and Eq. (15)

$$\boldsymbol{q}_1^{(i)} = \mathcal{H}(\boldsymbol{w}^{(i)}) , \tag{16}$$

$$\alpha_1^{(i)} = \frac{\|\boldsymbol{w}^{(i)}\|_{\ell 1}}{G} . \tag{17}$$

This results in a residual error $\boldsymbol{r}^{(i)}$ which captures the information lost in the coarse approximation

$$\boldsymbol{r}^{(i)} = \boldsymbol{w}^{(i)} - \alpha_1^{(i)}\boldsymbol{q}_1^{(i)} . \tag{18}$$

Next, to refine the approximation, a second 1-bit quantization is performed on this residual error $\boldsymbol{r}^{(i)}$. This step solves for the refinement term by applying the same strategy

$$\boldsymbol{q}_2^{(i)} = \mathcal{H}(\boldsymbol{r}^{(i)}) , \tag{19}$$

$$\alpha_2^{(i)} = \frac{\|\boldsymbol{r}^{(i)}\|_{\ell 1}}{G} . \tag{20}$$

Finally, the full 2-bit quantized approximation of the weight group, $\hat{\boldsymbol{w}}^{(i)}$, is obtained by combining the coarse approximation and the refined residual term, as defined in Eq. (9):

$$\hat{\boldsymbol{w}}^{(i)} = \alpha_1^{(i)}\boldsymbol{q}_1^{(i)} + \alpha_2^{(i)}\boldsymbol{q}_2^{(i)} . \tag{21}$$

This reconstructed $\hat{\boldsymbol{w}}^{(i)}$ is used in the forward pass of the network. Our quantization strategy provides an adaptive quantization lattice, facilitating adaptive mapping for ill-shaped weight distributions, as shown in Figure 2. R2Q fully exploits the limited 2-bit quantization space via two strategies: first, decomposing the 2-bit quantization problem into a 1-bit coarse estimate and a 1-bit residual refinement; second, ensuring the **distribution-independent optimal solution** of each 1-bit subproblem. The former enhances lattice utilization, while the latter ensures distribution-agnostic robustness.

**Backpropagation** The quantization process in R2Q involves the non-differentiable function $\mathcal{H}(\cdot)$, which blocks gradient flow during backpropagation. To circumvent this issue, we adopt the Straight-Through Estimator (STE) (Bengio et al., 2013), which approximates the gradient of the quantization function as the identity mapping. This allows gradients from the loss function $\mathcal{L}$ to propagate back to the full-precision weights $\mathbf{W}$. After R2Q quantizes and reconstructs $\mathbf{W}$, the backward propagation is formally expressed as

$$\frac{\partial\mathcal{L}}{\partial\mathbf{W}} = \frac{\partial\mathcal{L}}{\partial\hat{\mathbf{W}}}\frac{\partial\hat{\mathbf{W}}}{\partial\mathbf{W}} . \tag{22}$$

With STE, the quantizer's gradient is approximated as

$$\frac{\partial\hat{\mathbf{W}}}{\partial\mathbf{W}} \approx \mathbf{I} . \tag{23}$$

Substituting into Eq. (4), we obtain

$$\frac{\partial\mathcal{L}}{\partial\mathbf{W}} \approx \frac{\partial\mathcal{L}}{\partial\hat{\mathbf{W}}} . \tag{24}$$

This approximation yields a simplified and tractable gradient for updating the full-precision weights.

## 5 EXPERIMENTS

This section presents the experimental validation of the proposed R2Q. We first detail the experimental configuration, including implementation details, the synthetic data generation strategy, and evaluation benchmarks. We then present and analyze the results to demonstrate the performance of R2Q.

Table 1: Comparison of R2Q and other 2-bit quantization methods under coarse-grained (group size = -1) and fine-grained (group size = 128) settings. Evaluations are based on language understanding and modeling tasks. CR denotes the compression ratio of the model's memory usage. For ARC-c/e, BoolQ, Hellaswag, PIQA, and Winogrande, we report accuracy. For WikiText-2, we report PPL. R2Q employs two 1-bit kernels with separate scaling, doubling the scaling parameters under fine-grained quantization; thus, we set its group size to 256 (2 × 128) for parity.

| Model | Bit-Width | Method | Group-Size | CR | ARC-cc | ARC-e↑ | BoolQ↑ | Hella.↑ | PIQA↑ | Wino.↑ | MMLU↑ | Wiki2↓ |
|-------|-----------|--------|------------|-----|--------|--------|--------|---------|-------|--------|-------|--------|
| Llama-7B | bf16 | \ | \ | 100.00% | 41.98 | 75.38 | 75.17 | 56.95 | 78.78 | 69.46 | 31.33 | 9.39 |
| | 2-bit | LLM-QAT | -1 | 12.3% | 19.45 | 26.81 | 37.83 | 25.71 | 51.90 | 50.28 | 22.95 | 1776.98 |
| | | BitDistiller | -1 | 12.3% | 23.38 | 25.88 | 53.88 | 25.59 | 51.46 | 50.43 | **24.61** | 15938.61 |
| | | **R2Q (ours)** | -1 | 12.3% | **27.47** | **56.82** | **59.36** | **44.44** | **70.08** | **57.54** | 24.08 | **17.13** |
| | | LLM-QAT | 128 | 15.1% | 26.19 | 42.00 | 58.32 | 35.32 | 62.13 | 54.22 | 23.91 | 115.52 |
| | | BitDistiller | 128 | 15.1% | **28.67** | 57.91 | 63.18 | 41.76 | 69.10 | **60.22** | 24.26 | 33.51 |
| | | **R2Q (ours)** | 256 | 15.0% | 28.24 | **59.18** | **64.89** | **45.37** | **70.62** | 59.59 | **24.28** | **16.96** |
| OPT-6.7B | bf16 | \ | \ | 100.00% | 30.38 | 65.53 | 65.72 | 50.53 | 76.22 | 64.88 | 24.94 | 12.28 |
| | 2-bit | LLM-QAT | -1 | 12.5% | 19.03 | 38.51 | 60.64 | 30.34 | 60.39 | 50.59 | 22.92 | 53.24 |
| | | BitDistiller | -1 | 12.5% | 21.50 | 42.55 | 48.69 | 31.88 | 61.10 | 52.41 | 24.57 | 125.01 |
| | | **R2Q (ours)** | -1 | 12.5% | **25.68** | **58.08** | **63.7** | **44.19** | **71.43** | **60.77** | **24.60** | **16.71** |
| | | LLM-QAT | 128 | 13.5% | 19.71 | 39.23 | 59.97 | 36.17 | 61.75 | 51.01 | **25.63** | 31.07 |
| | | BitDistiller | 128 | 13.5% | **26.37** | 53.70 | 62.35 | 41.22 | 68.66 | 56.99 | 25.40 | 22.49 |
| | | **R2Q (ours)** | 256 | 13.4% | 24.32 | **58.96** | **64.40** | **44.31** | **72.58** | **60.54** | 25.09 | **16.81** |

**Experimental Setup.** Our experiments follow the knowledge distillation (KD) framework of LLM-QAT (Liu et al., 2023b), where the student model learns to approximate the teacher's outputs using Kullback–Leibler (KL) divergence loss. Training is conducted with the AdamW optimizer at a learning rate of $2 \times 10^{-5}$, zero weight decay, and a cosine learning rate schedule. The input sequence length is limited to 2048 tokens. We evaluate on widely adopted models, including OPT (Zhang et al., 2022), Llama (Touvron et al., 2023), and Qwen (Team, 2024; Yang et al., 2025). All experiments were executed on a single NVIDIA A800 GPU (80 GB), requiring approximately 6 GPU hours.

**Training Data.** Following the data-free paradigm of LLM-QAT, we generate synthetic training data using the teacher model. Each sequence is initiated with a uniformly sampled token, after which the teacher generates tokens autoregressively. To establish a coherent context, the initial 3–5 tokens are selected deterministically (top-1), while the rest are sampled stochastically to enhance diversity and mitigate mode collapse. We cap the sequence length at 512 tokens to balance diversity and cost.

**Evaluation Benchmarks and Metrics.** We evaluate our method on a suite of tasks for commonsense reasoning, question answering, and language modeling. For question answering and commonsense reasoning, we use accuracy as the primary metric. For language modeling, we report perplexity (PPL). In accordance with the current quantization methodology Du et al. (2024), bf16 precision is employed as our baseline. The benchmarks include **ARC** (Clark et al., 2018), which contains science questions at easy and challenge difficulty levels (ARC-e and ARC-C); **BoolQ** (Clark et al., 2019), a binary QA task based on paragraph entailment; and **HellaSwag** (Zellers et al., 2019), which tests commonsense inference by selecting the most plausible continuation. **PIQA** (Bisk et al., 2020) focuses on physical reasoning between two options, while **Winogrande** (Sakaguchi et al., 2021) evaluates pronoun disambiguation using nuanced contextual clues. **MMLU** (Hendrycks et al., 2020), evaluated under a 5-shot setting, covers a broad range of professional and academic subjects, serving as a comprehensive benchmark for measuring multi-domain knowledge and reasoning. For language modeling, we use **WikiText-2** (Merity et al., 2016), which assesses next-word prediction on Wikipedia text.

**Evaluation on Language Understanding and Modeling Tasks.** Table 1 presents a comprehensive evaluation of our proposed R2Q method. We evaluated R2Q against the current state-of-the-art 2-bit quantization method, BitDistiller Du et al. (2024). Consistent with BitDistiller's approach, we incorporated LLM-QAT Liu et al. (2023b) as a comparative method, which utilizes knowledge distillation. We report performance across seven representative benchmarks using two foundation models: Llama-7B (Touvron et al., 2023) and OPT-6.7B (Zhang et al., 2022). The compression ratio (CR) of the model's memory is presented.

For group-wise quantization, we consider coarse-grained and fine-grained settings. In coarse-grained settings, each channel forms a group (group size = -1). The fine-grained baseline uses a group

size of 128. Since R2Q uses two 1-bit kernels with separate scaling factors, it doubles the scaling parameters. To ensure a fair comparison with a similar parameter budget, we set R2Q's group size to 256. R2Q consistently outperforms baselines across all settings. In the coarse-grained setting, it yields substantial gains—for instance, on Llama-7B, R2Q boosts ARC-e from 26.81 (LLM-QAT) and 25.88 (BitDistiller) to 56.82, and reduces WikiText-2 perplexity from over 1700 and 15,938 to just 17.13. On OPT-6.7B, it improves HellaSwag accuracy from 30.34 to 44.19 and lowers perplexity from 53.24 to 16.71. In fine-grained settings, R2Q maintains its edge: with group size 256, it achieves top scores—58.96 on ARC-e and 72.58 on PIQA—and the lowest WikiText-2 perplexity (16.81), outperforming all baselines.

These results highlight R2Q's robustness across both discriminative and generative tasks. By decomposing quantization into two 1-bit kernels with enhanced error correction, R2Q effectively preserves model capacity under aggressive compression. The effectiveness of this design is confirmed through ablation studies based solely on the initial 1-bit coarse quantization. In addition, we provide a comparative analysis of quantization errors between R2Q and RTN. Further details are presented in Appendix C and Appendix D.

Table 2: Performance comparison between the original BitDistiller (BitDistiller (RTN)) and BitDistiller integrated with R2Q (BitDistiller (R2Q)) under coarse-grained (group size of -1) quantization.

| Model | Method | ARC-c↑ | ARC-e↑ | BoolQ↑ | Hella.↑ | PIQA↑ | Wino.↑ | MMLU↑ | Wiki2↓ |
|---|---|---|---|---|---|---|---|---|---|
| Llama-7B | BitDistiller (RTN) | **23.38** | 25.88 | 53.88 | 25.59 | 51.46 | 50.43 | **24.61** | 15938.61 |
| | BitDistiller (**R2Q**) | 20.99 | **40.11** | **54.80** | **30.17** | **60.66** | **51.46** | 23.05 | **310.03** |
| OPT-6.7B | BitDistiller (RTN) | 21.50 | 42.55 | 48.69 | 31.88 | 61.10 | 52.41 | 24.57 | 125.01 |
| | BitDistiller (**R2Q**) | **25.09** | **55.01** | **63.36** | **41.30** | **71.11** | **57.30** | **24.68** | **112.55** |
| Qwen2.5-7B | BitDistiller (RTN) | 20.39 | 23.76 | 42.20 | 25.02 | 52.45 | 49.64 | 26.85 | 42877.17 |
| | BitDistiller (**R2Q**) | **36.09** | **61.61** | **70.03** | **39.59** | **68.99** | **60.69** | **43.88** | **193.05** |
| Qwen3-4B | BitDistiller (RTN) | 20.39 | 26.30 | 38.10 | 25.75 | 52.34 | 48.86 | 24.48 | 44696.35 |
| | BitDistiller (**R2Q**) | **23.63** | **43.77** | **66.17** | **31.72** | **59.41** | **53.20** | **29.52** | **494.6** |

**R2Q as a Plug-and-Play Module.** To validate the versatility of R2Q, we integrate it into the BitDistiller framework under a coarse-grained quantization setting. Specifically, we compare BitDistiller using RTN (BitDistiller (RTN)) with its R2Q-enhanced counterpart (BitDistiller (R2Q)) across language understanding and modeling benchmarks on Llama-7B, OPT-6.7B, Qwen2.5-7B (Team, 2024) and Qwen3-4B (Yang et al., 2025) (Table 2). Further details can be found in Appendix B.

Across multiple models, R2Q consistently demonstrates superior generalization under aggressive quantization. On Llama-7B, while BitDistiller (RTN) performs slightly better on ARC-c, R2Q integration yields substantial improvements elsewhere, including +14.23 on ARC-e, +4.58 on HellaSwag, and a sharp reduction in WikiText-2 perplexity. On OPT-6.7B, R2Q consistently outperforms RTN, boosting BoolQ, PIQA, and Winogrande, achieving +12.46 on ARC-e, and lowering WikiText-2 perplexity from 125.01 to 112.55. On Qwen2.5-7B, the improvements are even more striking, with +37.85 on ARC-e , +27.83 on BoolQ, +17.03 on MMLU and a drastic drop in WikiText-2 perplexity. On Qwen3-4B, R2Q integration restores performance exceptionally well, achieving +28.07 on BoolQ , +17.47 on ARC-e, + on 5.04 on MMLU and reducing WikiText-2 perplexity from 44,696.35 to 494.6. These results highlight R2Q's remarkable effectiveness in recovering and enhancing model performance after extremely coarse-grained and low-bit quantization.

These results highlight R2Q's remarkable effectiveness in recovering and enhancing model performance after extremely coarse-grained and low-bit quantization, while maintaining or improving performance on challenging benchmarks such as MMLU.

**Quantization Stability.** Table 3 compares quantization stability under different group sizes, measured by the performance change ($\Delta$) across language understanding and modeling tasks. Smaller absolute values indicate better stability.

R2Q consistently outperforms BitDistiller and LLM-QAT across most tasks and group size transitions. For example, with Llama-7B from g256 to g-1, R2Q shows only -0.77% on ARC-c, versus -5.29% (BitDistiller) and -6.74% (LLM-QAT). On BoolQ, R2Q has a -5.53% drop, significantly lower than -9.3% and -20.49%. For OPT-6.7B, R2Q shows strong stability, even slight gains like +1.36% on ARC-c, and near-zero drops on HellaSwag (-0.12%) and WikiText-2 (-0.1%).

Table 3: Benchmark-wise comparison of quantization stability for Llama-7B and OPT-6.7B with varying group sizes.The table reports the changes in model performance from coarse-grained to fine-grained quantization. The smaller the absolute value, the more stable the quantization strategy.

| Model | Method | Group-Size | CR | ΔARC-c | ΔARC-e | ΔBoolQ | ΔHella. | ΔPIQA | ΔWino. | ΔMMLU | ΔWiki2 |
|---|---|---|---|---|---|---|---|---|---|---|---|
| | LLM-QAT | g128→g-1 | 15.1%→12.3% | -6.74 | -15.19 | -20.49 | -9.61 | -10.23 | -3.94 | -0.96 | 1661.46 |
| Llama-7B | BitDistiller | g128→g-1 | 15.1%→12.3% | -5.29 | -32.03 | -9.30 | -16.17 | -17.64 | -9.79 | 0.35 | 15905.10 |
| | **R2Q (ours)** | g256→g-1 | 15.0%→12.3% | **-0.77** | **-2.36** | **-5.53** | **-0.93** | **-0.54** | **-2.05** | **-0.20** | **0.17** |
| | LLM-QAT | g128→g-1 | 13.5%→12.5% | -0.68 | -0.72 | 0.67 | -5.83 | -1.36 | -0.42 | -2.71 | 22.17 |
| OPT-6.7B | BitDistiller | g128→g-1 | 13.5%→12.5% | -4.87 | -11.15 | -13.66 | -9.34 | -7.56 | -4.58 | -0.83 | 102.52 |
| | **R2Q (ours)** | g256→g-1 | 13.4%→12.5% | **1.36** | **-0.88** | **-0.70** | **-0.12** | **-1.15** | **0.23** | **-0.49** | **-0.10** |

These results validate R2Q effectively mitigates accuracy degradation during aggressive group-wise quantization, providing a robust choice for stable low-bit model compression.

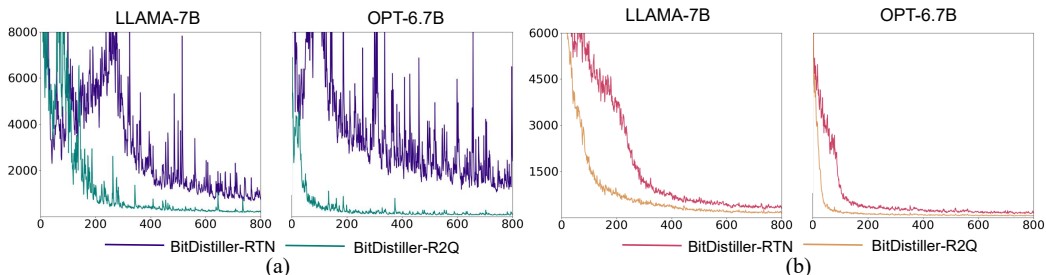

Figure 3: The (a) gradient norms and (b) training loss of BitDistiller (RTN) and BitDistiller (R2Q). The R2Q-integrated version (BitDistiller-R2Q) significantly reduces gradient fluctuations and converges faster and smoother.

**Gradient Stability and Convergence Efficiency.** Figure 3 shows training gradient norms for Llama-7B and OPT-6.7B. Standard RTN causes unstable, oscillatory gradients, while R2Q integration yields smoother curves and faster convergence, indicating better optimization dynamics and less quantization noise. These results show R2Q can effectively serve as a plug-in module, enhancing robustness and performance across architectures and tasks in coarse-grained quantization.

## 6 DISCUSSION AND FUTURE WORK

In this paper, we introduce R2Q, a residual refinement quantization framework. R2Q decomposes the 2-bit quantization problem into two optimal distribution-free 1-bit subproblems, offering greater flexibility than traditional strategies. Experiments confirm that R2Q enhances model performance across tasks, improves gradient stability, and accelerates convergence, especially in coarse-grained quantization. As a plug-and-play module, R2Q integrates into existing frameworks, delivering substantial gains without architectural changes.

**Limitations and Future Work.** While R2Q effectively tackles weight quantization, it shelves activation quantization, potentially limiting inference compression and speedup. Extending R2Q to joint weight-activation quantization is a key yet non-trivial direction. Another limitation lies in our heuristic assumption that combining two 1-bit kernels yields a distribution-independent optimal 2-bit quantization. Although empirically validated, complete theoretical proof remains to be explored. Future work includes developing high-performance R2Q operators implementable via 1-bit quantization logic (More details in Appendix E). Since the two 1-bit kernels in R2Q are independent, they can be computed in parallel without affecting inference efficiency. Applying R2Q to multimodal transformers may further test its generalizability and reveal modality-specific quantization behavior.

In summary, R2Q offers a flexible and generalizable foundation for advancing ultra-low-bit quantization of large-scale models, with strong potential for further innovation in both algorithmic and system-level design.

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
