# OpenReview forum: "R2Q: Residual Refinement Quantization for Robust 2-Bit Large Language Models"
_ICLR.cc/2026/Conference — ICLR 2026 Conference Withdrawn Submission_

### Official Review · Reviewer_9RB6 · 2025-10-26

**Soundness:** 2
**Presentation:** 2
**Contribution:** 3
**Rating:** 4
**Confidence:** 4

**Summary:**

This paper proposes an activation free, 2-bit quantization-aware training (QAT) method. The quantization method consists of two steps: 1 bit coarse approximation and 1 bit residual refinement. This quantization method is effective with QAT on many benchmarks.

**Strengths:**

1. This paper provides a novel 2-bit quantization method that decomposes a 2-bit quantization problem into 2 1-bit quantization subproblems, which brings performance gain on final benchmarks. They further show R2Q can be used in the QAT framework.
2. This paper touches on an interesting problem: the trade-off between the number of bits and the number of high-precision scalers given a fixed group size in group-wise quantization.

**Weaknesses:**

1. R2Q would increase the inference latency in theory because it brings the addition operation compared to 2-bit Round-to-Nearest, and the authors did not provide a numerical study of by how many degrees the slow down can be. To improve this, authors can provide speed profiling results and analyze that under what circumstances the latency is acceptable or give a practical guide about the latency.

2. Some benchmark improvement is meaningless because both R2Q and baseline methods are based on random guesses. For example, almost all MMLU numbers are based on random guesses, 25%, and the same for ARC-Challenge.

3. R2Q claims effectiveness on QAT, which consists of pretraining and supervised fine-tuning, but experiments are limited to QAT for continual pretraining. QAT for supervised fine-tuning (given a prompt, predicting the response) remains unexplored.

4. This paper does not compare R2Q to small models, which undermines the necessity of R2Q. The necessity of 2-bit QAT comes from the memory constraint of some devices. Therefore, it is natural to compare it with small models that take the same memory as quantized models. For example, a 2-bit Qwen3-8B and Qwen3-1.5B.

**Questions:**

1. What teacher model did you use in the experiments? Did you use one set of synthetic data for all model training, or does each model family have its own set of synthetic data? What is the training dataset size? How many epochs did you train? Please provide these necessary experiment details.
2. Why does LLM-QAT perform far worse than a random guess on ARC-C, only about 20%? I raise doubts about the soundness of the experiments.
3. Do you have the results of Qwen2.5-7B or Qwen3-8B in Table 1? Qwen is an important model family, and providing results of the Qwen series would increase the soundness of the experiments.
4. Can you compare 2-bit Qwen3-8B and Qwen3-1.5B? If 2-bit Qwen3-8B fails to match the performance of Qwen3-1.5B, R2Q is not attractive even if it outperforms other QAT baselines. I would raise my score to 6 if this concern can be addressed.

---

### Official Review · Reviewer_571J · 2025-10-27

**Soundness:** 2
**Presentation:** 1
**Contribution:** 2
**Rating:** 2
**Confidence:** 5

**Summary:**

To address the high memory consumption of large language models, the authors propose a 2-bit quantization-aware training (QAT) method that decomposes the 2-bit quantization task into two sequential 1-bit quantization subproblems.
For each quantization subproblem, the quantization parameter $\alpha$ is optimized by minimizing the quantization error.
Experimental results demonstrate that the proposed method improves the performance of 2-bit quantized models on language understanding and modeling tasks, compared with conventional uniform bit-width QAT methods.

**Strengths:**

1. The paper provides thorough mathematical derivations to justify the selection of quantization parameters, enhancing the rigor of the proposed approach.
2. The method is designed as a plug-and-play module, making it easily adaptable to various training loss functions and broadly applicable across different model architectures.

**Weaknesses:**

1. The performance of R2Q on certain benchmarks appears limited. For instance, in Table 1 and Table 2, the 2-bit LLaMA-7B model achieves around 25% accuracy on ARC-Challenge and MMLU, which is close to random choosing. This may suggest that further optimization is needed for challenging tasks.
2. The choice of baselines could be strengthened. For example, EfficientQAT [1], which employs a uniform bit-width QAT strategy, reports only a 5.35% performance drop on the 2-bit LLaMA2-7B model. In addition, the paper does not include comparisons with non-uniform quantization methods like QuIP# [2], which may limit the comprehensiveness of the experimental evaluation.
3. The paper focuses primarily on model accuracy and quantization methodology, but does not provide an assessment of hardware efficiency, such as latency and throughput. Including such results would help demonstrate the practical benefits of R2Q in real-world deployment scenarios.
4. Some notations and formulas in the paper could be presented more clearly. For example, in Equation (11), the notation for the Frobenius norm appears to be incorrect; in lines 247–248, the subscript $i$ is not properly formatted; and in Equations (13) and (15), the symbols $\mathcal{H}(\cdot)$ and $\text{sign}(\cdot)$ are used to denote the same operation. Refining these notations would improve readability and reduce potential ambiguity.

[1] Chen, Mengzhao, et al. "EfficientQAT: Efficient Quantization-Aware Training for Large Language Models."

[2] Tseng, Albert, et al. "QuIP$#$: Even Better LLM Quantization with Hadamard Incoherence and Lattice Codebooks." International Conference on Machine Learning. PMLR, 2024.

**Questions:**

See weaknesses.

---

### Official Review · Reviewer_6ayK · 2025-10-30

**Soundness:** 2
**Presentation:** 2
**Contribution:** 2
**Rating:** 2
**Confidence:** 4

**Summary:**

This paper proposes R2Q (Residual Refinement Quantization), a 2-bit quantization method for large language models that decomposes the quantization problem into two sequential 1-bit subproblems.
The authors claim R2Q enables an adaptive quantization lattice superior to static Round-To-Nearest (RTN) approaches and can be integrated as a plug-and-play module into existing quantization-aware training (QAT) frameworks.

**Strengths:**

**1. Conceptual novelty.** The core idea of decomposing 2-bit quantization into two 1-bit subproblems with residual refinement is interesting and provides a more adaptive quantization lattice compared to existing RTN-based methods.

**2. Comprehensive experimental evaluation.** The paper presents experiments across diverse benchmarks.
The ablation studies effectively demonstrate the contribution of the residual refinement stage.

**3. Improved training stability.** The gradient stability analysis in Figure 3 convincingly shows that R2Q integration reduces gradient fluctuations and enables faster, smoother convergence compared to RTN-based approaches.

**Weaknesses:**

**1. Poor performance despite QAT; weak empirical justification for 2-bit QAT usefulness.** Despite employing computationally expensive QAT, the 2-bit quantized models show severely degraded performance.
These substantial accuracy drops raise questions about whether 2-bit quantization with QAT is practically viable, especially when 4-bit PTQ methods like OPTQ and AWQ can achieve much smaller accuracy losses with significantly lower training costs.
Furthermore, comparison with 2-bit VQ methods is missing—these often perform well even with PTQ alone.
Although R2Q achieves fast inference, the resulting models perform too poorly to be of practical use, undermining the claimed motivation for ultra-low-bit quantization.

**2. Insufficient verification for mobile/edge deployment.** Although the motivation heavily emphasizes “on-device LLMs,” the paper does not present any runtime, latency, or memory measurements on actual mobile hardware or simulated platforms.
The claim that R2Q preserves “matmul-free efficiency” is discussed only theoretically in Appendix E, but it is never validated on real mobile devices or even on PC-level kernels.
At minimum, an implementation-level validation (e.g., using a standard CUDA or CPU kernel) is necessary to demonstrate that R2Q’s binary decomposition indeed provides computational benefits.
If embedded-device evaluation is infeasible, the authors should at least verify that the proposed kernel achieves stable runtime performance on a single commodity GPU under realistic inference workloads.
Without such evidence, the claim of deployability for resource-constrained environments remains speculative.

**3. Lack of scalability and architectural diversity.**
The experiments are limited to a small set of mid-sized models (OPT-6.7B, LLaMA-7B, and Qwen-4B/7B), leaving scalability unverified.
There is no evidence that R2Q generalizes to larger models such as 70B-scale LLMs, where quantization behavior often differs significantly due to wider activation distributions and heterogeneous layer norms.
In addition, the study omits evaluations on other architectures such as LLaMA-2, LLaMA-3, Qwen2, Mistral, and similar families, which are commonly used in the community.
Results on mixture-of-experts (MoE) models like Mixtral would also be essential to verify robustness under sparse-activation settings.
Without experiments covering diverse model sizes and architectures, the paper fails to demonstrate R2Q’s scalability and general applicability beyond the tested configurations.

**Misc.**
1. The paper suffers from numerous structural and formatting issues that hinder readability:
- Some citations are not properly written via ~\citep{} (e.g., lines 212, 372, 373).
- The appendix should be contained within the main manuscript after the references, following ICLR formatting rules. Placing it in a separate supplementary file disrupts readability.
- All tables and figures should appear at the top of each page within their corresponding sections for scientific clarity and consistency.
- There are many other structural inconsistencies throughout the manuscript, which collectively make the paper feel unpolished.
2. The released code is incomplete and non-reproducible - only a bare train.py and a few utility files are provided without README.md, example commands, or scripts (.sh) for reproducing the results.
Without a runnable setup, this cannot be considered a proper code release.
For rebuttal, the authors should ensure a complete and validated code release to avoid rejection.
3. From the reviewer’s viewpoint, the writing could be further improved.
Although the idea itself is simple and easy to understand, the writing lacks clarity and makes it unnecessarily difficult for the reviewer to follow the core message.

**Questions:**

Refer to Weaknesses

---

### Official Review · Reviewer_3Qko · 2025-10-31

**Soundness:** 1
**Presentation:** 1
**Contribution:** 1
**Rating:** 0
**Confidence:** 5

**Summary:**

This paper proposes R2Q, a new quantization scheme designed for ultra-low bitwidth quantization. However, the proposed method appears to be quite similar to existing quantization approaches [1, 2], and both experimental results and analyses are insufficient to demonstrate clear advantages. In my opinion, this paper requires substantial improvements before being considered in a top-tier conference.

**Strengths:**

Unfortunately, I did not identify any notable strengths in the current version of this paper.

**Weaknesses:**

* **Novelty.** The proposed method, R2Q, appears quite similar to previously published quantization schemes [1, 2]. The authors should properly cite related work and provide a detailed discussion highlighting the differences and advantages of R2Q.

* **Experimental setup.** The experiments primarily use outdated and relatively small models, e.g. Llama 7B and OPT 6.7B. This makes it difficult to assess whether the proposed method can be generalized to or remain effective for more recent and larger models.

* **Analysis.** Although the paper introduces a new quantization scheme, it lacks comparisons on key practical metrics, such as inference speed and memory usage against existing methods. These evaluations are critical for validating the efficiency of the approach.

**Questions:**

* Could you clarify the differences between R2Q and the quantization schemes used in existing methods [1, 2]?


**Reference**

[1] Xu, Chen, et al. "Alternating multi-bit quantization for recurrent neural networks." arXiv preprint arXiv:1802.00150 (2018).

[2] Kwon, Se Jung, et al. "Alphatuning: Quantization-aware parameter-efficient adaptation of large-scale pre-trained language models." arXiv preprint arXiv:2210.03858 (2022).

---

### Note · Authors · 2025-11-12

I have read and agree with the venue's withdrawal policy on behalf of myself and my co-authors.